# Thyroid Storm in Head and Neck Emergency Patients

**DOI:** 10.3390/jcm9113548

**Published:** 2020-11-04

**Authors:** Mohamed A. Radhi, Basaviah Natesh, Paul Stimpson, Jonathan Hughes, Francis Vaz, Raghav C. Dwivedi

**Affiliations:** 1Head and Neck Centre, University College London Hospitals NHS Trust, Euston Road, London NW1 2PG, UK; Paul.Stimpson@nhs.net (P.S.); Jonathan.Hughes@nhs.net (J.H.); Francis.Vaz1@nhs.net (F.V.); Raghav.Dwivedi@nhs.net (R.C.D.); 2Department of ENT and Head Neck Surgery, University Hospital Coventry and Warwickshire NHS Trust, Coventry CV2 2DX, UK; b.natesh@nhs.net

**Keywords:** thyroid storm, thyroid crisis, thyrotoxicosis, head neck surgery, head neck, head neck trauma, head neck foreign body, rapid sequence intubation, strangulation, suicide attempt

## Abstract

Background: Thyroid storm is a rare but life-threatening emergency that prompts urgent intervention to halt its potentially disastrous outcomes. There is not much literature available on thyroid storm in head neck trauma and non-thyroid/parathyroid head neck surgery. Due to rarity of thyroid storm in head and neck trauma/surgery patients, its diagnosis becomes challenging, is often misdiagnosed and causes delay in the diagnosis and management. Therefore, the aim of this work was to compile, analyze and present details to develop a consensus and augment available literature on thyroid storm in this group of patients. Materials and methods: A comprehensive literature search of the last 30 years was performed on PUBMED/MEDLINE, EMBASE, CINAHL and Science Citation Index for thyroid storm using MeSH words and statistical analyses were performed. Results: Seven articles describing seven cases of thyroid storm were reviewed. All patients required medical management and one patient (14.3%) required adjunctive surgical management. Burch and Wartofsky Diagnostic criteria for thyroid storm were used in diagnosis of 42% patients. Time of diagnosis varied from immediately upon presentation to formulating a retrospective diagnosis of having a full-blown thyroid storm at 4 days post presentation. It was misdiagnosed and unthought of initially in majority of these cases, (71.4%) were not diagnosed in the first day of hospital stay. Conclusion: Early recognition of thyroid storm in head and neck patients markedly reduce morbidity/mortality. Albeit unexpected, it should be ruled out in any symptomatic head and neck trauma or post-surgery patient.

## 1. Introduction

Thyroid storm (TS) or crisis is defined as an acute, decompensated state of thyroid hormone–induced, severe hypermetabolism involving multiple systems and is the most extreme state of thyrotoxicosis. It was first introduced in literature by Lahey [1] in 1928 as “The Crisis of Exophthalmic Goiter”. However, several cases were reported in patients without preexisting disease.

Based on national surveys from the United States the incidence of thyroid storm is 0.57 to 0.76 in 100,000 persons per year in patients ≥ 18 years of age with the mean age ± SE of 48.7 ± 0.11 years. It is more common in females as compared to males with the M: F of 1:3 and in-hospital mortality rates of 1.2–3.6% [2]. The Japanese experience reports a slightly lower incidence of 0.20/100,000 persons per year but much higher mortality rates of 10.7% [3].

TS manifests as an array of signs and symptoms affecting multiple organ systems including Central nervous system (CNS), Cardiovascular and Gastrointestinal/hepatic manifestations. Central nervous system (CNS) manifestations range from: Restlessness, delirium, mental aberration or psychosis, somnolence or lethargy and coma. Cardiovascular manifestations include tachycardia, atrial fibrillation and features of Congestive heart failure (CHF) namely: pulmonary edema, moist rales over more than half of the lung field and/or cardiogenic shock. Gastrointestinal (GI)/hepatic manifestations may include nausea, vomiting, diarrhea, or a total bilirubin level ≥ 3.0 mg/dL. Fever of ≥ 38 °C may well be a manifestation [4].

Thyroid storm is a manageable life-threatening emergency provided it is diagnosed and treated in a timely manner. It is common in patients with previous thyroid disorders and thus easily picked up and managed. The occurrence of thyroid storm being confined to patients with preexisting thyroid disease is debatable. However, patients with no previous history of thyroid disorders fall victim to such an unexpected disastrous outcome should it occur. Only a few cases of thyroid storm have been reported following trauma to the neck and only one case has been reported in a patient presenting with food impaction in the esophagus prompting urgent esophagoscopy [5].

The aim of this work is to augment available literature on thyroid storm in head and neck trauma and surgery patients without previous thyroid disease by reviewing all published cases in regard to clinical presentation, time to establish diagnosis, treatment and outcome.

## 2. Materials and Methods

### 2.1. Search Strategy and Inclusion Criteria

A comprehensive literature search of articles from the last 30 years on PUBMED/MEDLINE, EMBASE, CINAHL and Science Citation Index for thyroid storm in head neck trauma and head and neck surgery patients was performed. Key words used were thyroid storm; thyroid crisis; acute thyrotoxicosis; head and neck cancer/trauma/burn patients. The search was further extended by using other related MeSH words like ((“thyroid crisis” [MeSH Terms] OR (“thyroid” [All Fields] AND “crisis” [All Fields]) OR “thyroid crisis” [All Fields] OR (“thyroid” [All Fields] AND “storm” [All Fields]) OR “thyroid storm” [All Fields]) AND (“head and neck neoplasms” [MeSH Terms] OR (“head” [All Fields] AND “neck” [All Fields] AND “neoplasms” [All Fields]) OR “head and neck neoplasms” [All Fields] OR (“head” [All Fields] AND “neck” [All Fields] AND “cancer” [All Fields]) OR “head and neck cancer” [All Fields])) AND “humans” [MeSH Terms] which yielded 123 articles. MeSH words like ((“thyroid crisis” [MeSH Terms] OR (“thyroid” [All Fields] AND “crisis” [All Fields]) OR “thyroid crisis” [All Fields] OR (“thyroid” [All Fields] AND “storm” [All Fields]) OR “thyroid storm” [All Fields]) AND (“injuries” [Subheading] OR “injuries” [All Fields] OR “trauma” [All Fields] OR “wounds and injuries” [MeSH Terms] OR (“wounds” [All Fields] AND “injuries” [All Fields]) OR “wounds and injuries” [All Fields])) AND “humans” [MeSH Terms] which yielded 77 articles. Moreover, MeSH words like ((“thyroid crisis” [MeSH Terms] OR (“thyroid” [All Fields] AND “crisis” [All Fields]) OR “thyroid crisis” [All Fields] OR (“thyroid” [All Fields] AND “storm” [All Fields]) OR “thyroid storm” [All Fields]) AND (“Head Neck” [Journal] OR (“head” [All Fields] AND “and” [All Fields] AND “neck” [All Fields]) OR “head and neck” [All Fields])) AND “humans” [MeSH Terms] which yielded 11 articles. Finally MeSH words like ((“thyroid crisis” [MeSH Terms] OR (“thyroid” [All Fields] AND “crisis” [All Fields]) OR “thyroid crisis” [All Fields] OR (“thyroid” [All Fields] AND “storm” [All Fields]) OR “thyroid storm” [All Fields]) AND (“burns” [MeSH Terms] OR “burns” [All Fields])) AND “humans” [MeSH Terms] which yielded 4 articles.

Collectively 217 articles were reviewed and duplicates were removed (Figure 1). Publications on thyroid storm in head and neck surgery for: head and neck benign and malignant neoplasms, head and neck trauma, head and neck burns were included. Publications with previous thyroid/parathyroid disease, thyroid/parathyroid surgery, non head and neck burn/trauma patients were excluded from the review. Reference lists from the relevant articles were then inspected and cross-referenced and any other pertinent publications were added to the review. Relevant details of articles in languages other than English were extracted and tabulated by the authors separately. A total of seven articles (with seven cases) that satisfied the inclusion criteria, were tabulated and finally reviewed in this first-of-its-kind study.

### 2.2. Data Extraction and Analysis

Data on the year of publication, age and gender of the patients, initial presentation and time interval between presentation and diagnosis were recorded. Diagnostic modalities including subjective observations, imaging and diagnostic criteria were also noted (Table 1). Treatment given for thyroid storm and long-term outcome were also recorded. Descriptive analysis was performed and mean and median time interval between initial presentation and diagnosis (with 95% confidence intervals) were calculated (Table 2).

## 3. Results

Since the number of studies published on thyroid storm in non-thyroid/parathyroid head and neck patients is small, a proper meta-analysis was not possible.

The mean age (SD) of the patient population was 30.4 (SD = 10.0) and median age of 32 (range; 19–48) years with male: female ratio of 4:3. In majority of patients the initial presentation was trauma to the neck (6 pateints = 85.7%), blunt (5 patients = 71.4%) and sharp (1 patient = 14.3%). Causes of trauma ranged from assault (3 patients = 42.8%) to road traffic accidents, suicide attempts by hanging and accidental self-inflicted injury (1 patient = 14.3% in each category respectively). One patient (14.3%) developed thyroid storm following rapid sequence intubation [5].

Diagnostic criteria for Thyroid storm (Burch and Wartofsky score of ≥45) were used in making the diagnosis in 42% of patients. In one patient (14.3%) imaging modalities (CT and US findings) initiated the workup towards thyroid storm [5]. Incidental CT findings in 2 of the patients (28.5%) lead to the diagnosis of previously unknown Graves’ disease [6,10].

The mean time interval between initial presentation and diagnosis was two days and median 1.5 day, ranging from diagnosis on hospital day one to retrospective diagnosis of events on hospital day four. The main reason for delay in diagnosis and treatment was misdiagnosis. One patient (14.3%) was misdiagnosed with head injury (temporal contusion) as a cause of deteriorating GCS while another case (14.3%) was misdiagnosed with anoxic ischemic encephalopathy.

Most patients (85.7%) were treated for thyroid storm with beta blockers and antithyroid medication which resulted in dramatic clinical improvement within 24 h. Yet one patient (14.3%) was stabilized with inotropes and crystalloids. The diagnosis of thyroid storm was completely missed and was made in retrospect in this patient [4]. Norepinephrine was withdrawn with complete stabilization of hemodynamics two days after treatment initiation [5].

All patients survived and had a favorable recovery with treatment. Follow-up was organized for 28.5% cases with endocrinology for newly diagnosed Grave’s disease. Remaining 71.5% cases did not have any planned follow-up.

## 4. Discussion

A rare disease is defined by the European Union as one that affects less than 5 in 100,000 of the general population [12]. Based on national surveys in Japan, the incidence of thyroid storm is 0.2 per 100,000 annually and up to 0.76 per 100,000 in the United States [2]. This makes TS a rare disease by definition. Unrecognized and untreated thyroid storms can be fatal. Despite the advent of B-blockers, thyroid storm continues to have an associated mortality ranging between 10–30% [3,13].

Burch-Wartofsky Point Scale (BWPS) for Thyrotoxicosis (Table 3) is a predictor of the likelihood that biochemical thyrotoxicosis is thyroid storm and was laid out in 1993 by Burch et al. [14]. A BWPS ≥45 is highly suggestive of thyroid storm and rapid and aggressive multimodal management in ICU is recommended. In 2012, Akamizu et al. of Japan’s Thyroid Association (JTA) proposed new diagnostic criteria (Table 4) that were derived from BWPS for thyroid storm assigning two grades (TS1 & TS2) according to signs and symptoms. TS1 (definite) includes CNS plus one other manifestation or three manifestations other than CNS. TS2 (suspected) includes two manifestations other than CNS or history of thyroid disease, presents with exophthalmos and goiter, and meets either of the criteria for definite cases [2].

Three out of the seven cases (42.8%) reviewed utilized BWPS to aid in diagnosis. Yet the majority did not back their diagnosis by a set of diagnostic criteria. This indicates the unfamiliarity with this entity, delayed diagnosis and suboptimal management.

Thyroid storm can have varied clinical presentation including fever, tachycardia, hypertension, and neurological and GI abnormalities. It can also lead to hyperdynamic shock i.e., high output cardiac failure that can progress to hypotension and coma [15]. Patients reviewed in this article developed similar signs and symptoms yet presented with dissimilar complaints. A mandibular fracture, a food impaction, a head injury, a spear gun impacted in the neck, a suicide attempt and an assault all lead to thyroid storm. Wide range of presentation makes thyroid storm an imperative entity that needs to be ruled out as it’s consequences can be detrimental.

Management of thyroid storm entails tackling all of its pathophysiological effects. In addition to the above measures, the management often requires aggressive volume resuscitation to replace fluid losses from vomiting, diarrhea, and heightened insensible fluid loss. ß-Receptor Antagonists provide immediate management of troublesome tachyarrhythmias. This can be achieved by administering intravenous propranolol (1 mg every 5 min until the desired effect is achieved). Oral maintenance therapy (20 to 120 mg every 6 h) can be used until antithyroid drug therapy is effective.

Antithyroid Drugs are used to halt the effect of increased circulatory T3 and T4 and suppress thyroxine production. Methimazole is preferred to PTU because it causes a more rapid decline in serum thyroxine levels and has a lower incidence of serious side effects of (agranulocytosis). The initial dose of methimazole is 10 to 30 mg once a day, and the initial dose of PTU is 75 to 100 mg three times daily. The dose of both medications is reduced by 50% after 4 to 6 weeks of therapy.

To block the thyroxine release from thyroid gland, Iodine solutions, such as saturated solutions of potassium iodide (SSKI) or potassium iodide-iodine (Lugol’s solution—4 drops every 12 h) or intravenously as sodium iodide (500 to 1000 mg every 12 h) can be used. Lithium (300 mg orally every 8 h) is used as a substitute in patients with iodine allergy.

Thyroid storm can accelerate glucocorticoid metabolism and create a relative adrenal insufficiency. Therefore, in cases of thyroid storm associated with severe or refractory hypotension, hydrocortisone (300 mg IV as a loading dose, followed by 100 mg IV every 8 h) may help correct the hypotension [16]. Successful management of thyroid storm also requires treatment of the precipitating event which could be managed medically or surgically. Thyroid gland rupture and hematoma may warrant surgical exploration and evacuation of the hematoma [7].

The majority cases utilized beta blockers and thionamides as mainstay of their management. Intravenous steroids [10] and SSKI [6] were also used as adjuncts to treatment. In one case, hyperdynamic shock as a result of undiagnosed thyroid storm was managed by crystalloids and norepinephrine [5]. This lead to a longer treatment course and almost doubled the time for symptom remission from within 24 h to after 48 h of treatment initiation [5].

The diagnostic criteria proposed in the literature are sufficient, however as Akamizu says. “As the next obvious step, therapeutic procedures that aim for a better prognosis should be created” [2]. Thyroid storm had almost 100% mortality a century ago. With better understanding of its pathophysiology and management, the mortality rates have significantly reduced. However, they are still as high as 30% in cases presenting with predominant CNS symptoms and those with delayed diagnosis [17,18].

## 5. Conclusions

Prevalence of thyroid storm in non-thyroid head and neck trauma and surgical patients is low. However, it should be kept in mind especially in symptomatic patients presenting with trauma to the neck regardless of the mechanism. Prompt recognition of symptoms and signs, examination of thyroid gland and getting urgent Thyroid function tests (TFTs) may lead to an early diagnosis and treatment and an improved outcome in these patients. Since the prevalence of thyroid storm in non-thyroid head/neck trauma and surgical patients is low yet detrimental; any acute presentation to head and neck patient with deranged vital signs (HR/BP/Temp/RR) should alert a clinician to order TFTs.

## Figures and Tables

**Figure 1 jcm-09-03548-f001:**
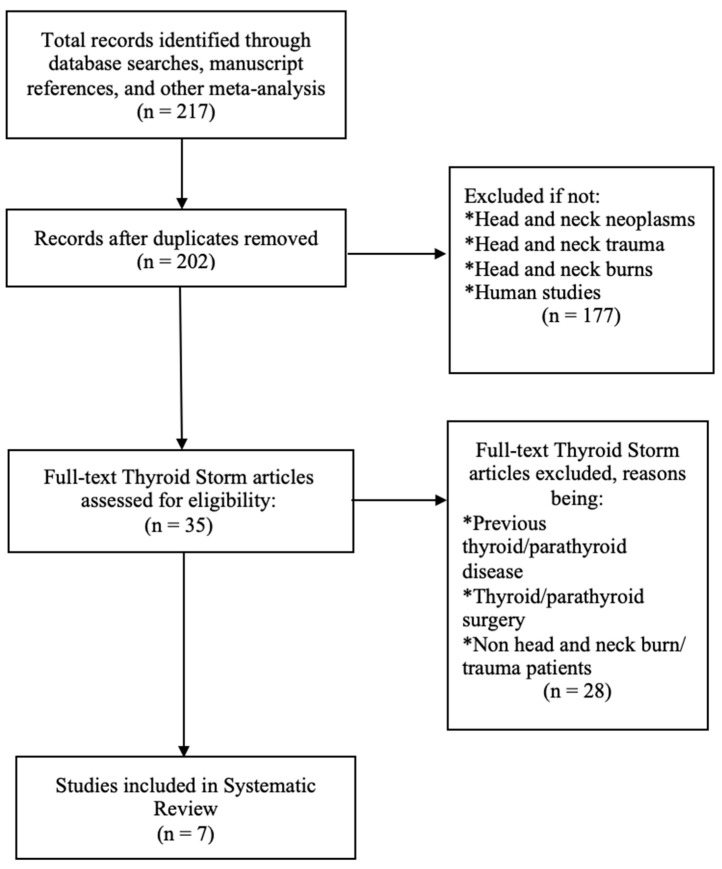
PRISMA flow diagram.

**Table 1 jcm-09-03548-t001:** Papers reviewed.

Paper	Age/Gender	Presentation	Comorbidities	Diagnostic Criteria	Time of Diagnosis	Management	Remission of Symptoms
Pride et al. [5]	48 yo/male	Food being stuck in his esophagus with associated PO in- tolerance.	Attention Deficit Hyperactive Disorder (ADHD) and hypertension	Burch and Wartofsky score 50	In retrospect, after stabilization in MICU hospital day 1	OGD deferred due to agitation upon sedation and RSI followed by stabilization and workup in MICU	Within 24 h
Yoshida et al. [6]	21 yo/ female	Assaulted, diffuse head and neck pain	-	low-grade fever 37.9, tachycardia 133 bpm, BP 165/94 mmHg, and a respiratory rate of 18 breaths/min., facial abrasions, swelling of the anterior neck, and a tremor	Hospital day 1	2.0 mg of intravenous dexamethasone, 300 mg of oral propylthiouracil (PTU), and a l.O-mg test dose of intravenous propranolol and SSKI.	Within 24 h
Hagiwara et. al. [7]	20 yo/Male	RTA, Mandibular fracture and cervical hematoma	-	Burch and Wartofsky score 50–65	In retrospect, hospital day 4	Continuous infusion of norepinephrine and crystalloids	After 48 h
Ramírez et al. [8]	37 yo/Female	Assaulted, found by paramedics with tourniquet tied around neck circumferential cervical contusion along with cervicofacial ecchymoses and petechiae.	-	Misdiagnosed initially as anoxic ischemic encephalopathy. A working diagnosis of thyroid stom made on basis of: Tachycardia, mental status changes, fever and deranged thyroid function test.	More than 72 h post presentation, hospital day 4	According to American thyroid association guidelines when diagnosis was established	Within 24 h
Delikoukos et al. [9]	32 yo/Male	Spear fishing-gun trident impacted in the right part of his neck	-	low-grade fever (37.9 °C), tachycardia (125 pulses/min), high blood pressure (165/105 mm Hg), facial abrasions and a tremor,	Hospital day 1	Medical treatment- unspecified prior to surgical neck exploration	Within 24 h
Karaören et al. [10]	36 yo/Male	Head injury following street fight, low GCS unexplained by minimal temporal lobe hematoma on CT	-	Burch and wartofsky score 70	24 h after ICU admission following stabilization in resuscitation and ICU	Propylthiouracil (PTU) 100 mg/4 h, propranolol 40 mg/8 h, and methylpredniso- lone 1 mg/kg and esmolol 50–100 μg/kg/min infusion.	Within 24 h
Shrum et al. [11]	19 yo/Female	Unsuccessful suicide attempt by hanging	-	Tachycardia, hypertension, delirium and hyperthermia along with imaging findings consistent with traumatic thyroid gland	More than 24 h after admission	Propranolol and methimazole	Within 24 h

**Table 2 jcm-09-03548-t002:** Showing means and medians for time interval between presentation and diagnosis (in days).

Number of Days	Mean	Median						
	Estimate	Std. error	95% confidence Interval		Estimate	Std. Error	95% confidence Interval	
			Lower bound	Upper bound			Lower bound	Upper bound
Ranging from hospital days 1–4	2.00	0.46	0.90	3.09	1.50	0.58	0.41	2.59

**Table 3 jcm-09-03548-t003:** Burch-Wartofsky Point Scale [14].

Temperature (F)	Cardiovascular Dysfunction	
99–99.9—5 points 100–100.9—10 101–101.9—15 102–102.9—20 103–103.9—25 ≥104.0—30	Tachycardia (beats/min)	
	99–109	5
	110–119	10
	120–129	15
	130–139	20
	>140	25
	Atrial fibrillation	10
Central Nervous system effects	Heart Failure	
Absent—0 points Mild (agitation)—10 Moderate (delirium, psychosis, extreme lethargy)—20 Severe (seizure, coma)—30	Mild (pedal edema)—5 Moderate (bibasilar rales)—10 Severe (pulmonary edema)—15	
Gastrointestinal-hepatic dysfunction	Precipitant History	
Moderate (diarrhea, nausea/vomiting, abdominal pain)—10	Positive—0
Severe (unexplained jaundice)—20	Negative—10	
Total: <25, storm unlikely; 25–45, impending storm; >45, thyroid storm		

**Table 4 jcm-09-03548-t004:** Japan thyroid association definition and diagnostic criteria for thyroid storm [4].

Prerequisite for Diagnosis: Presence of Thyrotoxicosis with Elevated Levels of Free Triiodothyronine (fT3) or Free Thyroxine (fT4).		
Diagnosis		
Grade of TS	Combinations of features	Requirements for diagnosis
TS1	First combination	Thyrotoxicosis plus at least one CNS manifestation and one of the following: fever, tachycardia, CHF, or GI/hepatic manifestation.
TS1	Alternate combination	Thyrotoxicosis and at least three of the following: fever, tachycardia, CHF, or GI/hepatic manifestations.
TS2	First combination	Thyrotoxicosis and a combination of two of the following: fever, tachycardia, CHF, or GI/hepatic manifestations.
TS2	Alternate combination	Meets the diagnostic criteria for TS1, except that serum fT3 or fT4 level are not available.
Exclusion and provisions		
Cases are excluded if other underlying diseases clearly causing any of the following symptoms: fever (e.g., pneumonia and malignant hyperthermia), impaired consciousness (e.g., psychiatric disorders and cerebrovascular disease), heart failure (e.g., acute myocardial infarction), and liver disorders (e.g., viral hepatitis and acute liver failure). Therefore, it is difficult to determine whether the symptom is caused by TS or is simply a manifestation of an underlying disease; the symptom should be regarded as being due to TS that is caused by these precipitating factors. Clinical judgment in this matter is required.		

TS, thyroid storm; TS1, “definite” TS; TS2, “suspected” TS.

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
