# Peer review of "Thyroid Storm in Head and Neck Emergency Patients"

_jcm, 2020, doi:10.3390/jcm9113548_

Round 1

Reviewer 1 Report

This is a review article "Thyroid storm in patients with head neck trauma and non-thyroid/parathyroid head neck surgery.  The authors did a literature search on online database and extracted data.   I have the following comments/suggestions.

  1. Line 47-54:  Need reference(s)
  2. Line 59-61:   However.....as an inpatient.  Too long sentence.  Consider revision.
  3. in materials/methods: please specify the interval of search.
  4. Line 94:   Where is figure 1?
  5. In results, since there are only 7 cases, please put actual numbers of patient followed by percentage.  
  6. Line 116-117: Causes of...    The 14.3% referred to which one, road traffic, suicide, or accidental, or???
  7. Line 118:   Reference#5 does not match with the case.
  8. Suggest to use median as the measurement of central tendency in all analysis of data(since there were only 7 subjects).
  9. Table 1:  Please include thyroid function results in all cases.
  10. Table 2:  Show only medians and related data.
  11. Line 272 -312:  Duplicated references?
  12. In results, also include the signs/symptoms that lead to the diagnosis in each patient.  This will be an importance information for readers to learn when to suspect thyroid storm in a specific clinical setting.
  13. In conclusions, it's too generalized.   Since the prevalence of thyroid storm in nonthyroid head/neck trauma and surgical patients is low, what symptoms/signs that could be more specific to alert a clinician to order TFTs.   What is the specific things that you want to point out after reviewing these cases.
  14. Suggest to get an adult endocrinologist in your institution to participate in this review.   This will provide more detail with more specific information/learning points from these cases.

Author Response

  1. Line 47-54:  Need reference(s)Table 4. Japan thyroid association definition and diagnostic criteria for thyroid storm
    Reviewer's comment Reply
    1- Line 47-54:  Need reference(s)  
    2- Line 59-61:   However.....as an inpatient.  Too long sentence.  Consider revision. "However, patients with no previous history of thyroid disorders fall victim to such an unexpected disastrous outcome should it occur. alongside various presentations to the emergency department and as an inpatient."

    3-in materials/methods: please specify the interval of search.

    " A comprehensive literature search of articles from the last 30 years"

    4-Line 94:   Where is figure 1?

    figure 1 added to supplementary material (PRISMA Chart)

    5- In results, since there are only 7 cases, please put actual numbers of patient followed by percentage.  

    Thank you, added.

    6- Line 116-117: Causes of...    The 14.3% referred to which one, road traffic, suicide, or accidental, or???

    Causes of trauma ranged from assault (3 patients= 42.8%) to road traffic accidents, suicide attempts by hanging and accidental self-inflicted injury (1 patient= 14.3% in each category respectively).

    7- Line 118:   Reference#5 does not match with the case.

    Apologies, corrected to reference 4 where needed.

    8- Suggest to use median as the measurement of central tendency in all analysis of data(since there were only 7 subjects).

    The mean age (SD) of the patient population was 30.4 (SD=10.0) and median age of 32.. The mean time interval between initial presentation and diagnosis was two days and median 1.5 day,

    9- Table 1:  Please include thyroid function results in all cases.

    unfortunately, not all the articles instated the TFTs.

    10- Table 2:  Show only medians and related data

    both mean and median were added to for clarity to ease comparison with national survey data as mentioned in introduction "Based on national surveys from the United States the incidence of thyroid storm is 0.57 to 0.76 in 100,000 persons per year in patients ≥18 years of age with the mean age +/-SE of 48.7 ± 0.11 years."
    11-Line 272 -312:  Duplicated references? Apologies, this has been corrected.

    12- In results, also include the signs/symptoms that lead to the diagnosis in each patient.  This will be an importance information for readers to learn when to suspect thyroid storm in a specific clinical setting.

    As Table 4 was added: Japan thyroid association definition and diagnostic criteria for thyroid storm, hopefully this will cover this domain.

    13- In conclusions, it's too generalized.   Since the prevalence of thyroid storm in nonthyroid head/neck trauma and surgical patients is low, what symptoms/signs that could be more specific to alert a clinician to order TFTs. What is the specific things that you want to point out after reviewing these cases.

    The following statement was added to the end of conclusion: Since the prevalence of thyroid storm in nonthyroid head/neck trauma and surgical patients is low yet detrimental; any acute presentation to head and neck patient with deranged vital signs (HR/BP/Temp/RR)should alert a clinician to order TFTs.

    14- Suggest to get an adult endocrinologist in your institution to participate in this review.   This will provide more detail with more specific information/learning points from these cases.

    A consideration of this suggestion will be taken into account, thank you.

Reviewer 2 Report

The review by Radhi M. et al is a review describing two main conditions of thyroid storm: patients with head and neck trauma and non-thyroid/parathyroid head and neck surgery. Although there are several review on "Thyroid storm" the originality of this paper is in these two conditions. However the significance is limited as the same authors stated that a proper meta-analysis was not possible since the number of studies was limited. For this reason I think that may be it is still early to program a meta-analysis on this topic.

Anyway, the review is still not in its final version as it seems that for example bibliography has not been completed. For example all the bibliography entries have to be placed in an appropriate numbering and style. 

The abbreviations have to be checked, too.

The table 3 and 4 are lacking.

There is confusion of capital letters and spaces that are lacking.

Finally, English language need a moderate revision.

Author Response

Reviewer's Comments Reply
The review by Radhi M. et al is a review describing two main conditions of thyroid storm: patients with head and neck trauma and non-thyroid/parathyroid head and neck surgery. Although there are several review on "Thyroid storm" the originality of this paper is in these two conditions. However the significance is limited as the same authors stated that a proper meta-analysis was not possible since the number of studies was limited. For this reason I think that may be it is still early to program a meta-analysis on this topic. The title of the review was changed:

Thyroid Storm in Head and Neck emergency patients

Anyway, the review is still not in its final version as it seems that for example bibliography has not been completed. For example all the bibliography entries have to be placed in an appropriate numbering and style. 

Thank you, the references have been updated.
The abbreviations have to be checked, too. Thank you, I have reviewed those to the best of my ability.
The table 3 and 4 are lacking. Apologies, these have been added. 
There is confusion of capital letters and spaces that are lacking. Thank you, I have reviewed those to the best of my ability.
Finally, English language need a moderate revision. Thank you, I have reviewed those to the best of my ability.

Reviewer 3 Report

In this paper the authors present results on thyroid storm for patients for trauma and non-thyroid/parathyroid head neck surgery.

The title is confusing.

The authors present a literature review for thyroid storm and head/neck trauma and non-thyroid/parathyroid head neck surgery and identify 7 papers.

The language used in the paper is confusing at times due to unusual phrasing and some claims are not supported with references – such as line 57 – ‘It is common in patients with previous thyroid disorders and thus easily picked up and managed.’ – how common do the authors claim thyroid storm is and what is their evidence for this claim?

I am unclear why the authors focused their search only on non-thyroid/parathyroid head and neck surgery and trauma? Do the authors feel that these patients are more likely to have thyroid storm? As such I do not understand why the authors have asked the research question and performed this study.

Figure 1 is not included in the manuscript.

There are multiple typos in the manuscript, line 94, line 97.. and the manuscript needs to be edited to an academic standard by the authors.

Table 1 is not readable and the results should be presented in another format.

The authors present a summary of seven case reports, six of which are trauma cases and one which is a food bolus – this questions the authors title and the introduction as from my reading no head/neck surgery patients are presented – surely a more appropriate title and introduction would be thyroid storm in head and neck emergency patients? I would suggest the paper be rewritten and resubmitted with this focus.

Table 2 the results should be presented as either mean or median depending on the distribution of the data, these results can also be reported in a simple sentence.

Author Response

Reviewer's Comments Reply

The title is confusing.

The authors present a literature review for thyroid storm and head/neck trauma and non-thyroid/parathyroid head neck surgery and identify 7 papers.

Thank you for your valuable comments!

The title has been changed: Thyroid Storm in Head and Neck emergency patients

The language used in the paper is confusing at times due to unusual phrasing and some claims are not supported with references – such as line 57 – ‘It is common in patients with previous thyroid disorders and thus easily picked up and managed.’ – how common do the authors claim thyroid storm is and what is their evidence for this claim?

Apologies for that, we will try to review it further to simplify the paper and make it less confusing.

The answer to the following question:

"how common do the authors claim thyroid storm is and what is their evidence for this claim?"

Will be found in the second paragraph of the introduction, line 40 onwards. The evidence being national surveys from the US and Japan. 

  • Lahey, Frank H. "The crisis of exophthalmic goiter." New England Journal of Medicine 1996 (1928): 255-257.
  • Galindo RJ1, Hurtado CR2, Pasquel FJ1, García Tome R2, Peng L3, Umpierrez GE1.“National Trends in Incidence, Mortality, and Clinical Outcomes of Patients Hospitalized for Thyrotoxicosis With and Without Thyroid Storm in the United States, 2004-2013.” Thyroid. 2019 Jan;29(1):36-43. doi: 10.1089/thy.2018.0275. Epub 2018 Dec 18.

Table 1 is not readable and the results should be presented in another format.

Table 1's format has been changed in terms of font, font size and spacing to enhance readability.

I am unclear why the authors focused their search only on non-thyroid/parathyroid head and neck surgery and trauma? Do the authors feel that these patients are more likely to have thyroid storm? As such I do not understand why the authors have asked the research question and performed this study.

The aim was to stress upon a detrimental yet easily missed aspect of thyroid pathology in previously euthyroid patients. The authors believe that the fact it has not been reported much doesn't mean it is uncommon and would rather be an indication of frequent misdiagnosis.

The following closing statement was added to the conclusion to recommend TFTs: "Since the prevalence of thyroid storm in non-thyroid head/neck trauma and surgical patients is low yet detrimental; any acute presentation to head and neck patient with deranged vital signs (HR/BP/Temp/RR) should alert a clinician to order TFTs."

Figure 1 is not included in the manuscript.

Apologies for this, it has been added.
There are multiple typos in the manuscript, line 94, line 97.. and the manuscript needs to be edited to an academic standard by the authors. Thank you, I have reviewed these a final spell check/ punctuation check/ capital small letter check, will be done throughout revision submissions.
The authors present a summary of seven case reports, six of which are trauma cases and one which is a food bolus – this questions the authors title and the introduction as from my reading no head/neck surgery patients are presented – surely a more appropriate title and introduction would be thyroid storm in head and neck emergency patients? I would suggest the paper be rewritten and resubmitted with this focus. The title has been changed: Thyroid Storm in Head and Neck emergency patients
Table 2 the results should be presented as either mean or median depending on the distribution of the data, these results can also be reported in a simple sentence. Both mean and median have been added to ease comparison of the former to data obtained from national survey's, and the later has been added as the sample size is 7 patients and median would be much rather appropriate.

Reviewer 4 Report

Radhi et al present here a nice overview of management of a very rare disease. This is an elaborate overview of this disease and is of help for colleagues doing literature research if such a case is presented in during their work. 

Minor:

A table with overview of the main findings would improve the aim of this work (including onset, side effects, duration, treatment options, outcome).

Author Response

Reviewer's Recommendation Reply
A table with overview of the main findings would improve the aim of this work (including onset, side effects, duration, treatment options, outcome).

Table 1 includes the following:

Age /Gender
Presentation
Comorbidities
Diagnostic criteria mentioned in each paper
Time of diagnosis since presentation
Management 
Outcome- (remission of symptoms)

Round 2

Reviewer 1 Report

The revised manuscript shows significant improvement.  Please double check the spelling again.  I still see some misspelling eg. line 209: detrimenta should be detrimental

Author Response

Comment:

The revised manuscript shows significant improvement.  Please double check the spelling again.  I still see some misspelling eg. line 209: detrimenta should be detrimental.

Reply:

Thanks for your valuable input. The article has been spell checked. Please consider the uploaded manuscript.

Reviewer 2 Report

In the present form this review is still unpublished.

Author Response

Comment:

In the present form this review is still unpublished. 

Reply:

Thanks for your valuable input. I have replied to all your comments previously and amended the article as advised. The article has been spell checked. Please consider the uploaded manuscript.

Reviewer 3 Report

The authors have responded appropriately to the peer reviewed comments and improved the manuscript, some of the phrasing is confusing but generally readable, I am unsure how much clinical interest and relevance this paper will have but the paper is much improved from its earlier version.

Author Response

Comment:

The authors have responded appropriately to the peer reviewed comments and improved the manuscript, some of the phrasing is confusing but generally readable, I am unsure how much clinical interest and relevance this paper will have but the paper is much improved from its earlier version.

Reply:

Thank you for your valuable input. Please consider the updated manuscript.